# Colorectal Polyp Size Classification Using a Siamese Network

**Author(s) names withheld**                                                                    EMAIL(S) WITHHELD

## Abstract

Colorectal cancer is one of the leading cause of cancer related deaths with increasing prevalence. One key factor in the likelihood of adenomatous cell differentiation is polyp diameter. There exist a significant cut-off value of 10 mm which clinicians use in diagnosis management. We propose a novel method to classify polyp size into above or below 10 mm classes based on a Siamese network. In a first step, a Siamese networks is trained to build a high dimensional feature embedding extracted for each polyp size. In as second step, we use a k-NN approach to classify polyp sizes based on the distance between the feature embedding of the input image, and the whole embedding space learned by the Siamese network. This method allows for better binary classification of the sub- and sup- 10 mm polyp size classes. Our data consist of around 55,000 images from 129 movies classified into various polyp sizes ranging from 1-15 mm. We trained our model on 10,746 images, and tested on 2,688 images equally split into each binary category. We obtained 79.2% in feature classification and 95.7% in polyp size classification.

**Keywords:** Computer Aided Diagnosis, Siamese Network, Polyp Classification

## 1. Introduction

Colorectal polyp size is a critical biomarker in colorectal cancer diagnosis and supervision, with larger polyp diameter having greater likelihood of adenocarcinomatous cell differentiation (Martínez et al., 2015; Klein et al., 2016; Summers, 2010). There are two significant thresholds in the clinical decision making process, which occur at 5 mm and 10 mm. There exists a high intra- and inter clinician variability in polyp size estimation (Elwir et al., 2017), and thus there is a need for an automatic classification system that can aid clinicians in their estimations. In this work we propose a binary classification model based on Siamese networks. By training a neural network to learn high dimensional descriptive features for each polyp category, that is, under or over 10 mm, we can classify new unseen instances of polyp images into the most relevant class. In a first step, we train a Siamese network to learn this embedding, and in a second step we use k-Nearest Neighbours (k-NN) to compute the closest class cluster to the query image. To the best of our knowledge, we present one of the first use of Siamese networks to classify polyp sizes. Furthermore, our method allows to build a high dimensional understanding of polyp features that ultimately can be used not only for binary classification.

Much research has been achieved on polyp classification. Most of them focus on polyp detection (Wang et al., 2018; Urban et al., 2018; Masashi et al., 2018; Zhang et al., 2017), although some works also focus on polyp characterization, such as histopathology (Korbar et al., 2017; Zhang et al., 2017). However, we have found few works that focus on polyp

size classification (Martínez et al., 2015; Itoh et al., 2019), which achieve the classification using depth estimation. As in-vivo colorectal ground truth depth data is difficult to obtain, the work in (Martínez et al., 2015) estimates depth using a technique know as defocus strategy, which requires blurring coefficients, trained using phantom data. The work in (Itoh et al., 2019) use pretrained monocular depth estimation neural networks, which they train using an unsupervised approach. However, the pretrained model was trained on computer vision based features and is therefore not adept at classifying medical imaging features. We propose a novel methodology using image information based on Siamese neural networks. Siamese neural networks were first used in (Bromley et al., 1993) to classify signatures. They consist of having two parallel neural networks which share weights between them. By optimizing over a similarity measure between the inputs, powerful discriminative features can be learned. This allows the network to generalize well to new, unseen data coming from unknown distributions, and as such have in recent years become popular in the field of computer vision for facial recognition tasks (Taigman et al., 2014; Koch et al., 2015; Varior et al., 2016), although they have been used in a few other applications (Baraldi et al., 2015; Bertinetto et al., 2016).

## 2. Methods

### 2.1. Data

Our data consist of 129 colonoscopic movies of different patients, each roughly lasting around 15 minutes. Experts annotated segments for each individual movie with the relevant polyp size and acquisition condition, such as white light, near blue infrared, and chromo. For each segment, the relevant movie frames where extracted into the appropriate polyp size class and saved as images. The data was then separated into above or below classes making up $4,478$ and $51,845$ images respectively. In order to have a balanced data set, we randomly selected $4,478$ images belonging to the below category. We split our data into training/testing datasets using a $80/20\%$ split. In order to train the Siamese network, we generated pairs of images classified into the same or different class. As such, our training data consisted of $10,746$ pairs of images of class: same above, same below, and different. Our testing data consisted of $2,688$ image pairs corresponding to the same classes.

### 2.2. Framework

We propose a Siamese based approach to obtain high dimensional feature embeddings such that images belonging to the same class have similar embeddings. The framework for our classification method is shown in Fig. 1. In a first step the Siamese network is trained by having pairs of images as inputs to a twin network that share weights. The network is based on the VGG16 architecture (Simonyan and Zisserman, 2014). The images are each embeded into a $4,096$ feature vector which are then fed as the input to an energy function. This function computes a metric that maps the distance between these high-level feature representations to be small if they are similar, and large if they are different. We use the contrastive loss function defined as:

$$\mathcal{L}(\mathbf{W}, Y, X_1, X_2) = (1 - Y)\frac{1}{2}(D_w(X_1, X_2))^2 + (Y)\frac{1}{2}\{max(0, m - D_w(X_1, X_2))\}^2, \quad (1)$$

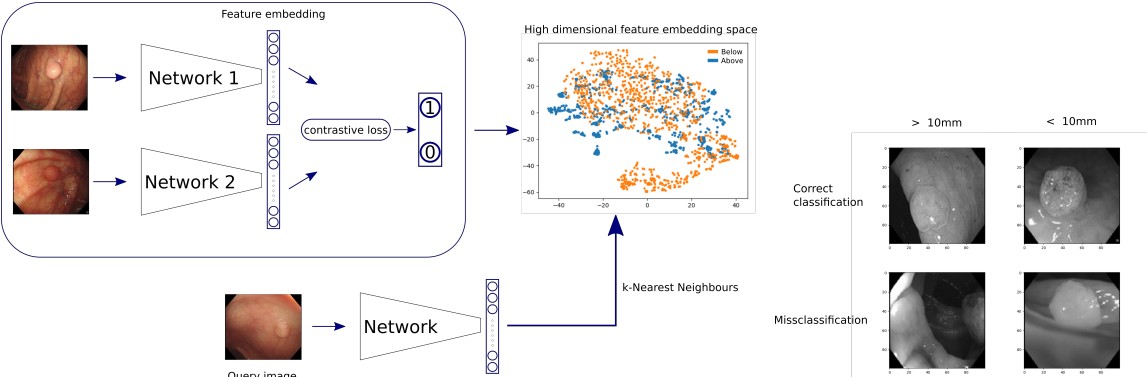

Figure 1: **Left:** Polyp classification framework. The Siamese network is shown at the the top left. Networks 1 and 2 represent twin networks with shared weights, based on the VGG16 architecture. New query images can be passed through the trained network and a kNN distance function is used to compute the output embedding to the rest of the feature embeddings. A T-SNE 2D representation of the embedding space can be seen at the top right. **Right:** Example of classification results.

where $\mathbf{W}$ represents the network weights, $X_1$ and $X_2$ represent the feature embeddings of the image pairs, with $Y = 1$ if they are similar, and $Y = 0$ if they are dissimilar. $D_w$ represents the $l_2$ norm, such that $D_w(X_1, X_2) = \|X_1 - X_2\|_2$, and $m$ is the margin. By training the network, a high dimensional feature embedding space can be obtained, as show in Fig. 1.

In a second step we use the trained network to compute the feature vector of a new query image and use the $k-$NN algorithm to compute the distance to the nearest class cluster.

## 3. Results and conclusion

We evaluated the Siamese network on $2,688$ image pairs belonging to the three classes mentioned in section. 2.1. This evaluated the discriminative power of the high-level embeding space learned by our network. Our network was able to classify whether pairs of images were similar or dissimilar with 70.2% accuracy. We then used the 1792 test images belonging to the above and below class to classify them using a $k$-NN approach. We were able to classify them with a 95.7% accuracy, using $k = 100$. In our experiments, each frame was randomly separated into training and testing data. Although there is clear separation between the training and testing data with regards to each frame, we need to consider different frames obtained from same video clip. As the test images also belong to the 129 movies, we believe that some classification bias could have occurred. We aim to obtain more movies to perform a more appropriate validation.

We presented a preliminary approach to polyp size classification based on Siamese network. While this work classifies the polyps into binary classes, our methodology can easily be adapted to work for the multi-class version of this problem.

## Acknowledgments

Acknowledgments withheld.

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
