# OpenReview forum: "Colorectal Polyp Size Classification Using a Siamese Network"
_MIDL.io/2019/Conference/Abstract — MIDL Abstract 2019_

### Official Review · AnonReviewer2 · 2019-04-27
**Colorectal polyp size classification method without a comparison baseline and without a solid validation**

**Rating:** 2
**Confidence:** 2

**Review:**

Paper presents a method for colorectal polyp size classification. The problem is posed as a two-step process with first embedding to lower-dimensional space done with Siamese networks followed by the classification with k-NN.
- Authors claim one of the first use of Siamese networks to classify polyp sizes. This may be true but not clear why would that be the best approach. In particular it is not clear why a direct supervised classification network was not trained. In fact, the method is not compared to any other approach or a baseline.
- As recognized by the authors the validation was not properly done as each frame was randomly separated into training and testing data without taking care that the frames from the same video end up in training or testing only.

---

### Official Review · AnonReviewer1 · 2019-05-01
**Binary Classification of Colorectal Polyp Size using Siamese embedding and KNN**

**Rating:** 3
**Confidence:** 2

**Review:**

The authors propose a method for binary classification polyp size into above or below 10 mm classes. Rather than using a standard classification loss (e.g., cross-entropy), they train  a Siamese network based on pairwise distances, so as to obtain a feature embedding. Then, the feature embedding is used in a KNN (K-nearest neighbour) approach to classify polyp sizes. This is interesting, and perhaps the first use of Siamese networks for the application at hand (in fact, I am not aware of many medical image analysis works based on Siamese networks). The authors mentioned that this pairwise-based embedding method, combined with KNN, yields better binary classification accuracy. I suppose they meant in comparison to standard CNN classifiers (e.g., based on cross-entropy loss). Please clarify? Also it will be nice to provide the results of a standard CNN classifier (to see the positive effect of this Siamese+KNN methodology). Any intuition why this Siamese embedding + KNN approach would be better than a standard cross-entropy classifier?

---

### Decision · Program_Chairs · 2019-05-06
**Acceptance Decision**

Accept